# Non-Alcoholic Steatohepatitis as a Risk Factor for Intrahepatic Cholangiocarcinoma and Its Prognostic Role

**DOI:** 10.3390/cancers12113182

**Published:** 2020-10-29

**Authors:** Stefania De Lorenzo, Francesco Tovoli, Alessandro Mazzotta, Francesco Vasuri, Julien Edeline, Deborah Malvi, Karim Boudjema, Matteo Renzulli, Heithem Jeddou, Antonietta D’Errico, Bruno Turlin, Matteo Cescon, Thomas Uguen, Alessandro Granito, Astrid Lièvre, Giovanni Brandi

**Affiliations:** 1Oncologia Medica, Azienda Ospedaliero-Universitaria di Bologna, via Albertoni 15, 40138 Bologna, Italy; stefania.delorenzo@studio.unibo.it; 2Division of Internal Medicine, Azienda Ospedaliero-Universitaria di Bologna, via Albertoni 15, 40138 Bologna, Italy; alessandro.granito@unibo.it; 3Service de Chirurgie Hépatobiliaire et Digestive, Centre Hospitalier Universitaire Pontchaillou Rennes, CIC-INSERM, Université de Rennes, 35000 Rennes, France; alex.mazzotta@gmail.com (A.M.); karim.boudjema@chu-rennes.fr (K.B.); heithem.jeddou@chu-rennes.fr (H.J.); 4Pathology Unit, S. Orsola-Malpighi Bologna Authority Hospital, 40138 Bologna, Italy; francesco.vasuri@aosp.bo.it (F.V.); deborah.malvi@aosp.bo.it (D.M.); antonietta.derrico@unibo.it (A.D.); 5Department of Medical Oncology, Centre Eugène Marquis, 35000 Rennes, France; j.edeline@rennes.unicancer.fr; 6Radiology Unit, S. Orsola-Malpighi Bologna Authority Hospital, 40138 Bologna, Italy; matteo.renzulli@aosp.bo.it; 7Service de Pathologie-Centre Hospitalier Universitaire Pontchaillou Rennes, INSERM Numecan U1241, Université de Rennes, Centre de Ressources Biologiques-BB-0033-00056, 35000 Rennes, France; Bruno.Turlin@chu-rennes.fr; 8Surgery Unit, Department of Medical and Surgical Sciences, University of Bologna, 40138 Bologna, Italy; matteo.cescon@unibo.it; 9Service de Hepatologie, Centre Hospitalier Universitaire Pontchaillou, 35000 Rennes, France; thomas.uguen@chu-rennes.fr; 10Department of Gastroenterology, Centre Hospitalier Universitaire Pontchaillou, University of Rennes, Inserm U1242, Rennes, France; astrid.lievre@chu-rennes.fr

**Keywords:** intrahepatic cholangiocarcinoma, non-alcoholic fatty liver disease, non-alcoholic steatohepatitis, outcome, liver cirrhosis

## Abstract

**Simple Summary:**

The prevalence of intrahepatic cholangiocarcinoma (iCCA) is rising. About 50% of iCCA arise in patients without known risk factors. We hypothesized that nonalcoholic fatty liver disease (NAFLD) and its most aggressive phenotype (NASH) could be risk factors for iCCA, similarly to other liver malignancies. We verified whether the prevalence of NAFLD and NASH was higher in the peritumor samples of iCCA patients compared with matched healthy controls (liver donors). We found the NASH (but not NAFLD) was over-represented in iCCA patients. Moreover, NASH patients had a shorter survival. Our results demonstrated that NASH is a risk factor for iCCA and underline the importance of dissecting the role of NASH from that of NAFLD as a whole. Prevention protocols for NASH patients should consider also the risk for iCCA and not only HCC. Studies aimed to find a direct pathogenic link between NASH and iCCA could add further relevant information.

**Abstract:**

Non-alcoholic fatty liver disease (NAFLD) and its most aggressive form, non-alcoholic steatohepatitis (NASH), are causing a rise in the prevalence of hepatocellular carcinoma. Data about NAFLD/NASH and intrahepatic cholangiocarcinoma (iCCA) are few and contradictory, coming from population registries that do not correctly distinguish between NAFLD and NASH. We evaluated the prevalence of NAFLD and NASH in peritumoral tissue of resected iCCA (*n* = 180) and in needle biopsies of matched liver donors. Data of iCCA patients were subsequently analysed to compare NASH-related iCCA (Group A), iCCA arisen in a healthy liver (Group B) or in patients with classical iCCA risk factors (Group C). NASH was found in 22.5% of 129 iCCA patients without known risk factors and in 6.2% of matched controls (risk ratio 3.625, 95% confidence interval 1.723–7.626, *p* < 0.001), while NAFLD was equally represented in both groups. The overall survival of NASH-related iCCA was inferior to that of patients with healthy liver (38.5 vs. 48.1 months, *p* = 0.003) and similar to that of patients with known risk factors (31.9 months, *p* = 0.948), regardless of liver fibrosis. The multivariable Cox regression confirmed NASH as a prognostic factor (hazard ratio 1.773, 95% confidence interval 1.156–2.718, *p* = 0.009). We concluded that NASH (but not NAFLD) is a risk factor for iCCA and might affect its prognosis. Dissecting NASH from NAFLD by histology is necessary to correctly assess the actual role of these conditions. Prevention protocols for NASH patients should also consider the risk for iCCA and not only HCC. Mechanistic studies aimed to find a direct pathogenic link between NASH and iCCA could add further relevant information.

## 1. Introduction

Intrahepatic cholangiocarcinoma (iCCA) is the second most frequent primary liver cancer following hepatocellular carcinoma (HCC) [1]. Recent epidemiological reports suggest an increasing worldwide incidence of both HCC and iCCA [1]. The widespread epidemic of non-alcoholic fatty liver disease (NAFLD) and its most aggressive form, non-alcoholic steatohepatitis (NASH), is held responsible for the increasing incidence of HCC [2]. On the contrary, the relationship between NAFLD and iCCA is still unclear, with conflicting data being reported [3,4]. The use of data deriving from population registries in which the diagnosis of NAFLD/NASH was searched using the International Classification of Diseases, Ninth Revision (ICD-9) code 571.8 did not discriminate between the two conditions, being both grouped under the same code and term (“Other chronic non-alcoholic liver disease”) [4,5,6]. In a more recent registry study, NAFLD was coded according to the results of the hepatic steatosis index, which cannot identify the subgroup of patients with NASH [7].

In the very few cases in which liver histology was systematically evaluated, the control group was not representative of the general population [8] or was not considered in the design of the study [9]. In the very recent World Health Organisation 5th edition classification of digestive system tumours, it has been acknowledged that iCCA is now better understood as an entity combining two different subtypes: A large duct type (which resembles extrahepatic cholangiocarcinoma) and a small duct type, which shares etiological, pathogenic, and imaging features with HCC (including metabolic syndrome and NAFLD) [10]. Still, it remains unclear whether NAFLD as a whole, rather than NASH alone, is actually a risk factor. Information about the prognostic role of metabolic risk factors is equally conflicting [9,11,12].

The definition of these aspects is of critical importance both for epidemiological and clinical reasons, especially considering the increasing prevalence of NAFLD and NASH. High-quality evidence is needed to provide reliable and useful information. For these reasons we designed a multicentre international study to assess whether: (1) NAFLD and NASH are over-represented in the peritumoral tissue of resected iCCA patients, using liver donors as a control group; (2) the presence of NAFLD/NASH influences the outcomes after surgical resection.

## 2. Results

A total of 180 patients with ICCA were included in this study (Figure 1), including 92 and 88 patients for the Italian and French centres, respectively. All of the included cases were “small-duct type” iCCA according to the World Health Organisation 5th edition classification [10].

### 2.1. Study Population

The clinical characteristics of the whole study population have been reported in Table 1.

Briefly, we included 98 (54.4%) male and 82 (45.6%) female patients, with a median age of 66 years (IQR 57–72) at the time of hepatectomy. Among the study population, 21 (14.7%) patients had known risk factors for iCCA. The median BMI was 25.1 kg/m^2^ (22.9–28.4), with a prevalence of hypertension, diabetes, and dyslipidemia of 47.2%, 17.8%, and 22.9%, respectively.

A majority of patients had a TNM stage Ia or Ib tumour at the time of liver resection, with a median main tumour dimension of 60 mm (IQR 38–86) (Table 2). The median follow-up was 32.5 months (IQR 15.4–60.1). The median OS was 44.6 months (95% confidence intervals 34.7–54.5).

### 2.2. Prevalence of NAFLD and NASH

Overall, 129 (86.3%) patients had no known risk for iCCA. Amongst them, 59 (45.7%) patients were classified as NAFLD and 29 (22.5%) as NASH, according to the FLIP algorithm. The prevalence of NAFLD (38.5 vs. 53.1%, *p* = 0.113) and NASH (23.4 vs. 21.5% *p* = 0.835) were similar in the Italian and French centres, respectively. 

In comparison, the prevalence of NAFLD and NASH was 38.8% and 6.2% amongst the controls, respectively. While the prevalence of NAFLD was similar in the two groups (*p* = 0.313), NASH was significantly over-expressed in iCCA patients (*p* < 0.001). Notably, only 11 iCCA patients (6.1% of the whole population) had a pre-operative medical history of NAFLD, and no patients had a diagnosis of NASH. 

Since the iCCA group had some patients with significant liver fibrosis (F3/F4), we performed an additional analysis to dissect the role of fibrosis from that of NASH. Adding fibrosis as a further matching factor was not feasible as too few patients with significant fibrosis were available amongst controls. As such, we considered only the 115 iCCA patients without significant fibrosis and performed a new matching with controls. In this analysis, our previous results were largely confirmed (Table 3). Notably, no patients had a pattern of fibrosis consistent with secondary biliary cirrhosis, due to a biliary obstruction by the tumour.

### 2.3. Characteristics of NASH-Related iCCA

As expected, NASH patients had a higher median BMI, a higher prevalence of hypertension and a trend toward a higher prevalence of diabetes mellitus compared to the remaining patients (Table 4). 

Regarding tumour characteristics, the main tumour size of NASH patients was slightly smaller than in patients with no known risk factors (50.0 vs. 65.5 mm), but similar to that of patients with classical risk factors. There were no differences in terms of multinodular disease, nodal involvement, rate of radical resection and adjuvant treatment. Amongst patients with a seemingly healthy liver (i.e., patients with no known risk factors for iCCA), we found four patients (4.0%) with advanced fibrosis (F3) and four patients (4.0%) with overt histological cirrhosis (F4), but no diagnostic criteria for known chronic liver disease. These patients were classified as cryptogenic advanced compensated liver disease. The rate of patients with significant fibrosis (F3–F4) decreased from a patients with classical risk factors (43.1%), to NASH patients (20.7%) and patients without risk factors (8.0%). At the pairwise analyses, there was a statistically significant difference between patients with classical risk factors and without risk factors, but not between NASH patients and any of the remaining groups, even if a trend was observed.

Since patients with NAFLD without a diagnosis of NASH (i.e., patients who are categorised as “non-alcoholic fatty liver” according to the current NAFLD guidelines [13]) were grouped together with patients with no steatosis in the Group B, we performed a further analysis to ensure that these two subgroups were not different in key prognostic features, possibly affecting the subsequent survival analysis. While the non-alcoholic fatty liver subgroup had an predictably higher prevalence of some metabolic risk factors in comparison with the subgroup with no steatosis [male sex 71.0 vs. 39.1% (*p* = 0.005); arterial hypertension 64.5 vs. 36.2% (*p* = 0.010); BMI 24.8 vs. 23.9 kg/m^2^ (*p* = 0.036)], the two populations were comparable in terms of key prognostic factors, including main tumour dimension (60 vs. 70 mm, *p* = 0.112), multinodular disease (29.0 vs. 23.2%, *p* = 0.619), *n* > 0 (16.1% vs. 26%, *p* = 0.217), R > 0 (19.4 vs. 36.2, *p* = 0.107), and access to adjuvant treatments (38.7 vs. 43.5%, *p* = 0.827).

### 2.4. Survival Correlates

Perioperative mortality was 1.7%, with one patient dying within 30 days in each of the three study groups. The median OS was 44.6 months (95%CI 34.7–54.5), with no differences between the Italian and French centres (*p* = 0.715). Main tumour dimensions, multinodular disease, resection margins, sex, and NASH were significantly associated with the OS at the univariate analysis. 

The multivariable Cox regression analysis confirmed tumour dimension, multinodularity, resection margins, classical risk factors and NASH as independent predictors of survival (Table 5). The sensitivity analysis reported an E-value of 2.333 for the association between NASH and OS. The limit of the 95% CI closest to the null hypothesis was 1.580. These values mean that the observed association between OS and NASH could be nullified only by an unmeasured confounder associated with both OS and NASH, with a strength similar to that of multinodularity, which is unlikely to have been missed. On the contrary, a weaker confounder could not disprove this association.

Stratifying the patients according to their underlying liver condition, the median OS of NASH patients was inferior to that of patients without risk factors (38.5 vs. 48.1 months, *p* = 0.003) and similar to that of patients with classical risk factors (31.9 months, *p* = 0.948) (Figure 2).

On the contrary, the TTR was similar in the three groups (NASH: 17.1 months (95% CI 15.9–18.2); no risk factors: 20.6 months (95% CI 6.6–34.6); classical risk factors 19.9 months (95% CI 9.6–30.2), *p* = 0.604)). The sub-analysis of the group of patients with no risk factors found no differences in terms of OS (*p* = 0.236) and TTR (*p* = 0.991) between patients with non-alcoholic fatty liver and no steatosis, respectively.

## 3. Discussion

In our study, we evaluated the role of NASH both as a risk and as a prognostic factor for iCCA. We report two substantial results. First, NASH (but not NAFLD) was over-represented in patients with resected iCCA in comparison to healthy controls. Second, NASH was associated with a shorted OS amongst patients without classical risk factors for iCCA.

Regarding the first point, the crude prevalence of NASH was 22.5% amongst patients without classical risk factors for iCCA and 16.1% amongst the whole study population. On the contrary, NAFLD was equally represented in both groups, strongly highlining the importance of a histological analysis to dissect the effects of this more aggressive form of liver steatosis from NAFLD as a whole (a kind of information that registry study is likely to miss). The prevalence of NASH in our iCCA group was similar to that previously reported in the only two studies systematically assessing liver histology [8,9]. In comparison with previous papers, however, our study combined the strengths of a multicentre approach, and matching for the main demographic and metabolic risk factors. Moreover, the sensitivity analysis showed that only very strong confounders may explain away our results. 

Overall, our results point toward a possible pathogenic role of NASH in iCCA. The pathogenic pathway leading from NASH to iCCA is still unclear [14]. In the case of NASH-related HCC, intracellular lipid accumulation with potential lipotoxicity [15,16], alterations in the control of cell cycle and apoptosis [17], disruption of the peroxisome proliferator-activated receptor-γ (PPAR-g) coactivators [18] have been all proposed as factors enhancing the inflammatory damage and potentially contributing to the liver remodelling and fibrosis. Moreover, disruptions in the gut microbiota [19,20,21] and specific polymorphisms of the PNPLA3 genes have been advocated are possible contributors to hepatocarcinogenic [22]. Future mechanistic studies investigating whether NASH-related HCC and iCCA have common (or, rather, different) pathogenic pathways are needed in the next future. From an epidemiological point of view, our data suggest that NASH might (at least partially) justify the increasing worldwide incidence of iCCA, whose reasons have remained unclear so far [10]. Therefore, we suggest that future cost-effectiveness analyses of possible primary and secondary prevention strategies of NASH-related liver cancers take into account not only NASH-related HCC, but also iCCA.

Regarding the second main finding of our study (namely the characteristics and the outcome of NASH-related iCCA), we provided novel evidence. As expected, the typical NASH patient had more features of metabolic syndrome than patients with classical risk factors or without risk factors at all. Most NASH-related iCCA developed in the liver with F0–F2 fibrosis, differently from a patient with viral hepatitis. This trend mirrors the well-known characteristics of NASH-related HCC, which can develop in the noncirrhotic liver in up to 40% of the cases [2,23,24]. The main tumour dimension was significantly smaller in the NASH group in comparison with patients with no risk factors, but similar to that in patients with known risk factors. As a small part of cirrhotic NASH patients (and most viral and alcoholic cirrhosis) were under surveillance for HCC, this difference could be partially due to this confounding factor, rather than actually representing a specific biological characteristic of the tumour [25,26]. 

Finally, we found a higher risk of death in the NASH group. The increased risk of death of NASH patients could theoretically derive from a different biological aggressiveness of NASH-related iCCA, the underlying liver disease or comorbidities. A direct response cannot be provided by our study, as it was impossible to ascertain the exact cause of death of every single patient without breaching the current privacy policies for observational studies. Despite this important limitation of our study, many data can help in exploring all of the possible hypothesis. The hypothesis of a different biological aggressiveness of NASH-related iCCA did not find clear support, for at least three different reasons. First, the tumour staging at the diagnosis was similar in the three groups. Second, the TTR was also similar between the study groups. Third, the survival curves of NASH and patients without risk factors groups overtly opened after 36 months, a time after which tumour-related deaths became less likely. The hypothesis of the role of the underlying liver disease would seem appealing and intuitive, as severe fibrosis and cirrhosis are concurrent causes of death in NASH-related HCC [2]. Cumulating evidence suggest that cirrhosis itself does not impact on survival of iCCA patients [25,26,27,28]. Consistent with these data, severe fibrosis was not related to the OS in our study. Thus, it is unlikely that severe fibrosis and cirrhosis alone justify the different OS. Comorbidities are another possible (and the most likely) explanation of different survivals. NASH patients, in particular, are well-known to be affected by concomitant cardiovascular medical conditions. In our study, both patients with no risk factors and NASH had a low prevalence of comorbidities, as reflected by the very low peri-operatory mortality. However, Hobeika and colleagues [11] recently reported that resected iCCA patients with metabolic syndrome had worst postoperative outcomes. Notably, the rate of NASH was also higher in patients with metabolic syndrome (25 vs. 5.4%, *p* = 0.005) [11]. 

Our study has some limitations. We already mentioned the impossibility to ascertain the exact cause of death of our patients. Moreover, it should be noted that our data have been collected in two high-volume centres for iCCA, limited to resected patients. This approach was needed to obtain a systematic confirmation of NASH and achieve the goal of recruiting a large number of patients in a limited period of time (to avoid confounders, such as improvement in surgical techniques and therapies for the underlying liver disorders). However, as a consequence, caution is needed before extending our results about prognosis to patients resected in low volume centres or in unresectable cases.

## 4. Materials and Methods

### 4.1. Study Setting and Design

Our study involved two large European Centres (Bologna Authority Hospital S.Orsola-Malpighi (Bologna, Italy), and Centre Hospitalier Universitaire Pontchaillou (Rennes, France)).

The primary aim of this study was to compare the prevalence of histology-confirmed NAFLD and NASH in patients with resected iCCA and controls between 2007 and 2016. The secondary aim was to compare the characteristics and outcomes of NASH-related iCCA with those of patients with classical risk factors for iCCA or with no risk factors at all.

Therefore, we retrospectively analysed prospectively collected data of consecutive patients diagnosed with iCCA and resected in our Institutions between January 2006 and December 2017. Patients were followed-up until 31 October 2019.

For the primary aim of our study, resected iCCA patients with no classical risk factors (including HBV and HCV infections, alcohol consumption > 20 g for women and 30 g for men, professional exposure to asbestos or chemicals) acted as cases. No patients had received steatogenic drugs (such as steroids or chemotherapies) in the last two years. Liver donors, whose biopsies had been performed in the same timeframe, acted as controls. Cases and controls were matched for age, sex, diabetes mellitus, and BMI in a 1:1 fashion. Liver donor cases were also matched for each institution.

For the secondary aim, iCCA patients with no previously known risk factors were re-categorised as NASH (Group A) or no risk factors (Group B) and compared with patients with classical risk factors (Group C). Data about demographics, comorbidities, clinical and pathological tumour characteristics were available for each patient (including tumour size and number, vascular and/or perineural invasion, nodal status, resection margins, and underlying liver fibrosis). The AJCC 8th edition staging system was adopted to re-stage iCCA according to the most recent classification [29]. 

### 4.2. Evaluation of NAFLD and NASH

The peritumoral tissues (for cases) and the liver needle biopsies (for controls) were examined for the presence of NAFLD and NASH. Surgical specimens and biopsies were analysed by trained pathologists with at least a 10-year experience in the liver histology, blind each other and to clinical data. Discordant cases were collegially re-evaluated.

The evaluation of the histology samples was performed as recommended by the Fatty liver inhibition of progression (FLIP) consortium [30]. For each specimen, a Steatosis, Activity, and Fibrosis (SAF) score, summarising the main histological lesions, was defined [30]. This assessed both and separately the grade of steatosis (S), the grade of activity (A), and the stage of fibrosis (F), the latter according to the NASH Clinical Research Network (CRN) [30,31]. Steatosis was assessed by the percentage of hepatocytes containing large and medium-sized intracytoplasmic lipid droplets (but not foamy microvesicles), on a scale of 0 to 3 (S0: <5%; S1: 5–33%, S2: 34–66%, S3: >67%). Ballooning of hepatocytes was graded from 0 to 2 (0: normal hepatocytes; 1: presence of clusters of hepatocytes with a rounded shape and pale cytoplasm, usually reticulated; 2, as for grade 1, but with at least one enlarged ballooned hepatocyte (at least 2-fold size compared with that of normal cells within a cluster of hepatocytes with grade 1 ballooning). Lobular inflammation was defined as a focus of two or more inflammatory cells within the lobule organized either as microgranulomas or located within the sinusoids. Foci were counted at 20 × magnification (grade 0: none; 1: < 2 foci per lobule; 2: >2 foci per lobule). The grade of activity (A from A0 to A4) was calculated by the addition of grades of ballooning and lobular inflammation [30]. 

Fibrosis was staged as follows: stage 0, none; stage 1, perisinusoidal or periportal fibrosis (stage 1a, mild perisinusoidal; stage 1b, moderate perisinusoidal; stage 1c, portal/periportal); stage 2, perisinusoidal and portal/periportal fibrosis; stage 3, bridging fibrosis; and stage 4, cirrhosis [30,31]. Severe fibrosis was defined as bridging fibrosis or cirrhosis [30].

A diagnosis of NAFLD was based on: (i) A history of no or limited daily alcohol intake (<20 g for women and <30 g for men); (ii) the presence of hepatic steatosis confirmed by histological examination; and (iii) exclusion of other liver diseases [13]. 

The diagnosis of NASH required the combination of three histological features: Steatosis, ballooning/clarification of hepatocytes, and lobular inflammation, according to a definition which has progressively gained acceptance in the liver community [32]. The FLIP algorithm [30] was used for the diagnosis of NASH. Steatosis was used as the criterion for entry into the algorithm weighted by hepatocellular ballooning and lobular inflammation. A case presenting with at least grade 1 of each of the three features (steatosis, ballooning, lobular inflammation) was classified as NASH [30].

As a final step, we performed a sensitivity analysis to verify how strong an unmeasured confounder would have to be to disprove the possibly different prevalence of NASH in both groups.

### 4.3. Evaluation of the Clinical Characteristics of NASH-Related iCCA

All of the iCCA patients were subsequently stratified according to their NASH status.

Patients with NASH were compared with patients with classical risk factors and patients with no risk factors in terms of demographics (age, sex), clinical characteristics (BMI, diabetes, arterial hypertension), liver fibrosis, and tumour characteristics (maximum tumour dimension, multinodularity, nodal involvement, TNM staging, resection margins, subsequent adjuvant chemotherapy).

### 4.4. Evaluation of the Prognostic Role of NASH

Perioperative mortality, time to recurrence (TTR) and overall survival (OS) of the three groups were analysed.

Perioperative mortality definition was death within 30 days from the surgical intervention. TTR was defined as the interval from surgery to tumour recurrence. OS was considered from the day of surgery to the day of death or the most recent follow up visit. A sensitivity analysis was performed to evaluate how strong an unmeasured confounder would have to be to disprove the possibly observed relationship between the study group and OS.

### 4.5. Statistical Analysis

Categorical variables were expressed as frequencies. Continuous variables were presented as median (range). Group matching was performed using the “Case Control Matching” function of SPSS version 23.0 (SPSS Inc., Chicago, IL, USA), which includes the Python Essential functions. A perfect match was required for sex and diabetes mellitus, the tolerance for continuous variables was set as ± 2 years for age and ± 1 kg/m^2^ for BMI. Three-group comparisons were subsequently performed with the Kruskal–Wallis test for continuous variables and the Pearson Chi-square test for categorical variables. Differences were considered significant at *p* < 0.05. In case of significant difference, post-hoc tests were performed for the pairwise comparisons. In particular, the Dunn-Bonferroni test and the Chi-square test for independence with a Bonferroni correction for multiple comparisons were used for continuous and categorical variables, respectively. 

OS and TTR were reported as median values and expressed in months, with 95% confidence intervals (CI). Survival curves were estimated using the product-limit method of Kaplan Meier. The role of stratification factors was analysed with log-rank tests. Variables reaching a *p* < 0.10 at the univariate analysis were further examined by backward stepwise multivariate analysis. Multivariate regression analysis was performed with the Cox hazards model. Moreover, in this case, differences were considered significant at *p* < 0.05. The sensitivity analysis was performed calculating the E-value and the limit of its 95% CI closest to the null [33,34]. All of the statistical analyses were performed with the same statistical software used for group matching.

### 4.6. Ethics

This study was approved by the Independent Ethics Committee of both participating centres. All the patients who were still living at the beginning of this study provided written informed consent. Given the retrospective analysis of the data, the Ethics committees lifted the necessity of informed consent for all the patients who had died before the start of the study-related analyses. All methods were performed in accordance with the relevant guidelines and regulations.

## 5. Conclusions

In conclusion, we found that NASH acts both as a risk factor and a prognostic factor for iCCA. In particular, NASH was found in more than 20% of patients who would have been classified as having no classical risk factors. In about 80% of cases, NASH-related iCCA arose in livers without severe fibrosis. The OS of NASH-related iCCA is similar to that of patients with viral or alcoholic liver disease and significantly lower than that of patients without any known risk factor for iCCA. Prevention protocols for NASH patients should also consider the risk for iCCA and not only HCC. Mechanistic studies that aim to find a direct pathogenic link between NASH and iCCA could add further relevant information.

## Figures and Tables

**Figure 1 cancers-12-03182-f001:**
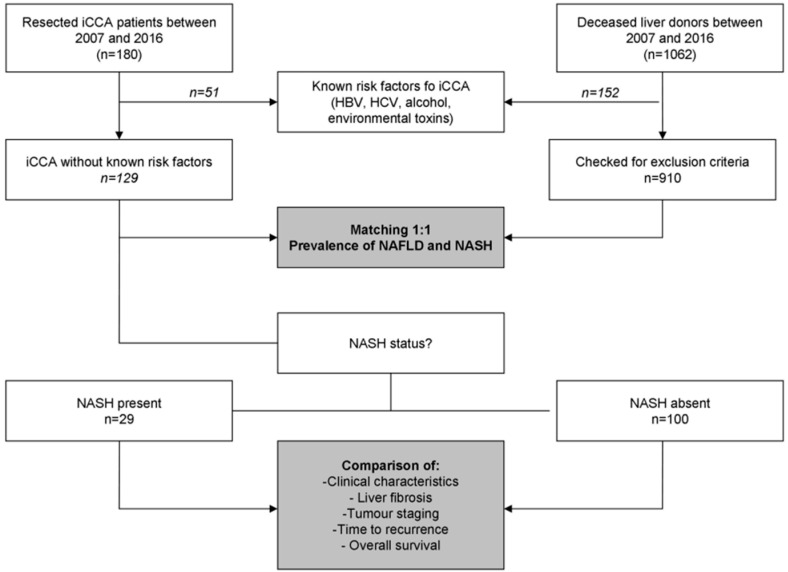
Patients disposition. iCCA, intrahepatic cholangiocarcinoma; NAFLD, non-alcoholic fatty liver disease; NASH, non-alcoholic steatohepatitis.

**Figure 2 cancers-12-03182-f002:**
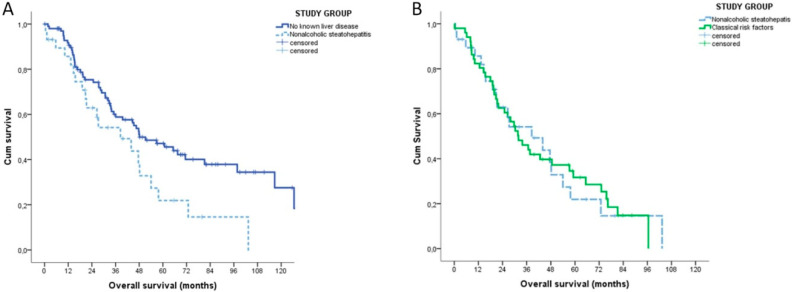
Kaplan-Meyer overall survival analysis of NASH patients (*n* = 29). (**A**) Patients without any known causes of liver disease (*n* = 100). (**B**) Patients with classical risk factors for intrahepatic cholangiocarcinoma (*n* = 51).

**Table 1 cancers-12-03182-t001:** Clinical characteristics of the whole study population.

Patients Characteristics (*n* = 180)	Value
Male sex	98 (54.4%)
Median age (years)	66 (57–72)
History of other tumours	38 (21.1%)
Arterial hypertension	85 (47.2%)
Type 2 diabetes mellitus	35 (19.4%)
Dyslipidemia	41 (22.8%)
Body mass index (median)	25.1 (22.9–28.4)
Risk factors	
Hepatitis B virus infection	8 (4.4%)
Hepatitis C virus infection	15 (8.3%)
Alcohol consumption	17 (9.4%)
Other *	11 (6.1%)
No classical risk factors	129 (71.7%)

* Primary sclerosing cirrhosis (*n* = 4), professional exposure to asbestos or chemicals (*n* = 4), Wilson disease (*n* = 2); hemochromatosis (*n* = 1).

**Table 2 cancers-12-03182-t002:** Tumour staging in the whole study population. Continuous variables have been reported as median (interquartile range).

Characteristics of Tumour (*n* = 180)	Value
Main tumour dimension (mm)	60 (38–86)
Multinodular disease	50 (27.8)
T	
1a	38 (21.1)
1b	48 (26.7)
2	66 (36.7)
3	13 (7.2)
4	15 (8.3)
N1	36 (20.0)
TNM stage (8th edition)	
Ia	37 (20.6)
Ib	42 (23.3)
II	51 (28.3)
IIIa	7 (3.9)
IIIb	43 (23.9)
Resection margin	
R0	131 (72.8)
R1	46 (25.6)
R2	3 (1.7)
Adjuvant treatment	76 (42.2)
Overall Survival (months, 95% CI)	44.6 (34.7–54.5)

Categorical variables are reported as frequencies (percentages).

**Table 3 cancers-12-03182-t003:** Prevalence of non-alcoholic fatty liver disease (NAFLD) and non-alcoholic steatohepatitis (NASH) in patients with resected intrahepatic cholangiocarcinoma (iCCA) without classical risk factors and liver donors.

Group	iCCA Patients	Liver Donors	*p*	Risk Ratio (95% CI)	E-Value (95% CI Closest to the Null)
**All patients (F0–F4)**					
NAFLD	59/129 (45.7)	50/129 (38.8)	0.313	1.18 (0.89–1.57)	1.64 (1.00)
NASH	29/129 (22.5)	8/129 (6.2)	<0.001	3.63 (1.72–7.63)	6.71 (2.84)
**No significant fibrosis (F0–F2)**					
NAFLD	53/115 (46.1)	49/115 (42.6)	0.690	1.08 (0.80–1.46)	1.38 (1.00)
NASH	21/115 (18.2)	5/115(4.3)	0.001	4.20 (1.64–10.76)	7.87 (2.66)

**Table 4 cancers-12-03182-t004:** Comparison of the clinical characteristic of iCCA patients with classical risk factors, non-alcoholic steatohepatitis, and no risk factors. Continuous variables have been reported as median (interquartile range). Categorical variables are reported as frequencies (percentages).

Variables	NASH Group A (*n* = 29)	No Risk Factors Group B (*n* = 100)	Classical Risk Factors Group C (*n* = 51)	Omnibus *p*-Value *	A vs. B *p*-Value **	A vs. C *p*-Value **
**Sex (Male)**	14 (48.3%)	49 (49.0%)	35 (68.6)	0.056	1.000	0.096
**Age (years)**	70.0 (58.0–75.5)	66.5 (58.0–72.8)	63.0 (56.0–70.0)	0.120	0.287	-
**History of other tumours**	7 (24.1%)	21 (21.0%)	10 (19.6)	0.891	0.799	0.777
**Hypertension**	22 (75.9%)	45 (45.0%)	28 (54.9)	0.013	0.005	0.092
**Diabetes**	7 (24.1%)	13 (13.0%)	14 (27.5)	0.073	0.085	0.798
**Dyslipidemia**	8 (27.6%)	22 (22.0%)	11 (21.6)	0.795	0.618	0.591
**Body mass index**	29.8 (24.1–32.0)	24.2 (22.0–26.7)	26.1 (23.6–28.3)	<0.001	0.001	0.030
**Severe fibrosis (F3–F4)**	6 (20.7)	8 (8.0)	22 (43.1)	<0.001	0.084	0.053
**Main tumour dimension (mm)**	50.0 (30.5–79.0)	65.5 (50.0–95.0)	50.0 (30.0–70.0)	0.003	0.005	1.000
**Multinodular disease**	10 (34.5%)	25 (25.0%)	15 (29.4)	0.576	0.346	0.802
**T**						
1a	5 (17.2%)	17 (17.0%)	16 (31.4)			
1b	4 (13.8%)	34 (34.0%)	10 (19.6)			
2	15 (51.7%)	31 (31.0%)	17 (33.0)	0.163	0.511	0.438
3a	3 (10.3%)	8 (8.0%)	2 (3.9)			
3b	2 (6.9%)	10 (10.0%)	6 (11.8)			
***n > 0***	28(27.6%)	23 (23.0%)	5 (9.8)	0.085	0.626	0.058
**R > 0**	9 (31.0%)	31 (31.0%)	9 (17.6)	0.193	1.000	0.178
**Adjuvant treatment**	12 (41.4%)	42 (42.0%)	22 (43.1)	0.986	1.000	1.000

* Three-group comparisons were performed with the Kruskal–Wallis test for continuous variables and the Pearson Chi-square test for categorical variables. ** The Dunn-Bonferroni test and the Chi-square test for independence with a Bonferroni correction for multiple comparisons were used for the subsequent pairwise comparison of continuous and categorical variables, respectively.

**Table 5 cancers-12-03182-t005:** Univariate and multivariate Cox regression analysis of overall survival in the whole study population (*n* = 180).

Univariate			Variable	Multivariate		
Hazard Ratio	95% CI	*p*-Value		Hazard Ratio	95% CI	*p*-Value
**0.674**	0.426–1.065	0.091	Sex (Female)	0.628	0.402–1.012	0.056
**1.011**	0.994–1.029	0.207	Age			
**1.005**	0.920–1.098	0.915	Centre (France)			
**1.014**	0.979–1.051	0.434	Body mass index			
**1.152**	0.709–1.872	0.567	Diabetes			
**Ref.**	Ref.	Ref.	No risk factors	Ref.	Ref.	Ref.
**1.809**	1.077–3.040	0.025	Classical risk factors	1.900	1.025–3.209	0.016
**1.724**	1.134–2.621	0.011	NASH	1.773	1.156–2.718	0.009
**1.307**	0.836–2.043	0.252	Severe fibrosis			
**1.008**	1.003–1.014	0.002	Main tumour dimension (mm)	1.010	1.004–1.015	<0.001
**2.105**	1.425–3.110	0.011	Multinodularity	2.043	1.375–3.036	0.028
**1.385**	0.896–2.140	0.142	*n* > 0			
**1.526**	1.017–2.290	0.041	R > 0	1.596	1.055–2.416	0.033
**0.909**	0.624–1.323	0.617	Adjuvant treatment			

NASH, non-alcoholic steatohepatitis.

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
