# Peer review of "Non-Alcoholic Steatohepatitis as a Risk Factor for Intrahepatic Cholangiocarcinoma and Its Prognostic Role"

_cancers, 2020, doi:10.3390/cancers12113182_

Round 1

Reviewer 1 Report

1)  General comments

Intrahepatic cholangiocarcinoma (iCCA) is an aggressive malignancy with poor outcome. As indicated in this article, the relationship between NAFLD and iCCA is still unclear. In this paper, authors concluded that NASH, not NAFLD, is a risk factor for iCCA and might affect its prognosis.

The purpose of this study and the conclusions in this manuscript are interesting and worthy of note. The study design and methods used in this article are adequate but not sufficient for the purpose of this study. 

 I would like to recommend authors to reply to the following comments.

2)  Specific comments

a) Major

(1)  Thirty cases of NAFLD excluding NASH are included in no risk factor group and used for analysis of clinical characteristics and overall survival in Table 4, Table 5, and Figure 2. Analysis of non-NASH NAFLD group or no risk factor group excluding non-NASH NAFLD cases is  needed, because concordance rates among pathologists for evaluation of NAFLD/NASH are currently poor.

Author Response

REVIEWER 1

1)  General comments

Intrahepatic cholangiocarcinoma (iCCA) is an aggressive malignancy with poor outcome. As indicated in this article, the relationship between NAFLD and iCCA is still unclear. In this paper, authors concluded that NASH, not NAFLD, is a risk factor for iCCA and might affect its prognosis.

The purpose of this study and the conclusions in this manuscript are interesting and worthy of note. The study design and methods used in this article are adequate but not sufficient for the purpose of this study. 

 I would like to recommend authors to reply to the following comments.

2)  Specific comments

  1. a) Major

(1)  Thirty cases of NAFLD excluding NASH are included in no risk factor group and used for the analysis of clinical characteristics and overall survival in Table 4, Table 5, and Figure 2. Analysis of non-NASH NAFLD group or no risk factor group excluding non-NASH NAFLD cases is  needed because concordance rates among pathologists for evaluation of NAFLD/NASH are currently poor.

RESPONSE: We thank the reviewer for this important methodological comment. We totally agree with this suggestion. We compared a comparison of “non-NASH NAFLD cases” and patients without steatosis, i.e the two kind of patients which had been aggregated as “No known risk factors”. The reviewer will find that “non-NASH NAFLD cases” have been called  “non-alcoholic fatty liver (NAFL)” in the text, as suggested by the current NAFLD guidelines.  This comparison find no differences in terms of key prognostic factors between the two subgroups, thus suggesting that the “No known risk factors” group was quite homogeneous. Indeed, some minor differences were noted in terms of metabolic risk factors, but this result was predictable and did not affect our findings as none of these metabolic alteration were found related with survival. As a consequence, no OS or TTR differences were found between non-NASH NAFLD subject and patients without steatosis. We added the following lines to the text (green font):

Line 151: Since patients with NAFLD without a diagnosis of NASH (i.e. patients who are categorized as “non-alcoholic fatty liver” according to the current NAFLD guidelines) were grouped together with patients with no steatosis in the Group B, we performed a further analysis to ensure that these two subgroups were not different in key prognostic features, possibly affecting the subsequent survival analysis. While the non-alcoholic fatty liver subgroup had an predictably higher prevalence of some metabolic risk factors in comparison with the subgroup with no steatosis [male sex 71.0 vs 39.1% (p=0.005); arterial hypertension 64.5 vs 36.2% (p=0.010); BMI 24.8 vs 23.9 Kg/m2 (p=0.036)], the two populations were comparable in terms of key prognostic factors, including main tumour dimension (60 vs 70 mm, p=0.112), multinodular disease (29.0 vs 23.2%, p=0.619), N>0 (16.1% vs 26%, p=0.217), R>0 (19.4 vs 36.2, p=0.107), and access to adjuvant treatments (38.7 vs 43.5%, p=0.827).

Line 185: The subanalysis of the group of patients with no risk factors found no differences in terms of OS (p=0.236) and TTR (p=0.991) between patients with non-alcoholic fatty liver and no steatosis, respectively. 

 ************THE REVIEWER’S COMMENTS SEEMS TO BE TRUNCATED. WE WERE NOT ABLE TO SEE ANY OTHER POINTS. WE ARE AT FULL DISPOSAL OF THE REVIEWER IN CASE OF FURTHER POINTS NEEDING TO BE ADDRESSED ***********

Reviewer 2 Report

In this manuscript, the authors demonstrate that NASH is a risk factor for iCCA and that the presence of NASH in the background liver influences post-operative outcome in iCCA patients. Overall, the manuscript is well-structured, the goals are clearly stated and the overall message is concise and interesting. However, I do have a few questions/comments.

1) Although the risk factors for iCCA are not as "well-established" as for HCC, metabolic syndrome/NAFLD is actually listed as one of the risk factors for iCCA, including in the recent WHO classification. I think that this study is interesting in that it focuses on NAFLD as risk factors for iCCA; however, it appears in this manuscript as if NAFLD/metabolic syndrome is NOT a known risk factor for iCCA.

2) It is not stated whether these iCCA cases were all "peripheral-type" iCCAs - were there any iCCAs that were located in/around the larger bile ducts towards the 1st/2nd order branches? The "known" risk factors for iCCA tend to differ depending on whether they are small duct or large duct type iCCAs. Were any cases associated with fluke infestation or stones?

3) Did any of the cases with F3/F4 fibrosis demonstrate a biliary-type pattern of fibrosis/cirrhosis, which could have arisen in association with biliary obstruction by the tumor?

4) Were there any differences in the clinicopathological characteristics between the Italian and French cohort? How many cases from each cohort were enrolled in the study? Were the donor liver cases also matched for each institution?

5) The authors conclude that "NASH was found in more than 20% of patients who would have been classified as having a healthy liver". Did any of these patients have clinical features of metabolic syndrome (hypertension etc) or liver function tests suggestive of NAFLD/NASH, or were they all diagnosed as NAFLD/NASH post-operatively after examining the background liver? The quoted sentence in the conclusion seems to imply that they were all incidentally discovered after surgery, but I feel that a good proportion of them would have actually been diagnosed with metabolic syndrome or NAFLD pre-operatively. 

6) There are few places with awkward phrasing and typographical errors in the manuscript, for example:

Line 32: "hyperexpressed" in iCCA patients

Line 92: A majority of patients "hade"

Line 225: "cause fo death"

Line 335: "an healthy liver"

Author Response

In this manuscript, the authors demonstrate that NASH is a risk factor for iCCA and that the presence of NASH in the background liver influences post-operative outcome in iCCA patients. Overall, the manuscript is well-structured, the goals are clearly stated and the overall message is concise and interesting. However, I do have a few questions/comments.

  • Although the risk factors for iCCA are not as "well-established" as for HCC, metabolic syndrome/NAFLD is actually listed as one of the risk factors for iCCA, including in the recent WHO classification. I think that this study is interesting in that it focuses on NAFLD as risk factors for iCCA; however, it appears in this manuscript as if NAFLD/metabolic syndrome is NOT a known risk factor for iCCA.

The reviewer raises a very interesting and relevant point (also referring to point 2). When our study was designed, the classification of iCCA referred to the WHO 2015 classification. In 2019, a new WHO classification was published. According to this update, iCCA should be divided into two distinct subtypes. The “small duct type” originates from the distal biliary branches and shares many risk factors with HCC (including metabolic syndrome), while the “large duct type” originates form 1st/2nd order branches and shares risk factors with the distal and peri-hilar cholangiocarcinoma (for instance flukes and stones). While the whole NAFLD has been referenced as a risk factors in the WHO classification, our study demonstrated that NASH alone is over-represented in iCCA. According to the reviewer’s suggestion, we modified the text as follows (blue font):

Line 75: “In the very recent World Health Organisation 5th edition classification of digestive system tumours, it has been acknowledged that iCCA is now better understood as an entity combining two different subtypes: a large duct type (which resembles  extrahepatic cholangiocarcinoma) and a small duct type, which shares etiological, pathogenic, and imaging features with HCC (including metabolic syndrome and NAFLD) [10]. Still, it remains unclear whether NAFLD as a whole, rather than NASH alone, is actually a risk factor”

  • It is not stated whether these iCCA cases were all "peripheral-type" iCCAs - were there any iCCAs that were located in/around the larger bile ducts towards the 1st/2nd order branches? The "known" risk factors for iCCA tend to differ depending on whether they are small duct or large duct type iCCAs. Were any cases associated with fluke infestation or stones?

From the previous point, this is a very interesting and rightful observation. Our study started in 2018, when the WHO 2015 classification was in use. However, even at that time we chose not to include in this study  12 cases in which the tumour was located in/around the 1st/2nd order branched with imaging and pathology features which did not allow a clear distinction between intrahepatic and peri-hilar cholangiocarcinoma. Following the publication of the new WHO classification, we confirmed our choice to exclude these cases. Also, we re-evaluated the tumour histology of the included cases and we can confirm that all of our patients had a “small duct type” iCCA. We added the following specification to the text:

Line 90: All of the included cases were “small-duct type” iCCA according to the World Health Organisation 5th edition classification.

  • Did any of the cases with F3/F4 fibrosis demonstrate a biliary-type pattern of fibrosis/cirrhosis, which could have arisen in association with biliary obstruction by the tumor?

We agree on the relevance of this point. No cases of F3-F4 fibrosis were characteristic of cholestatic-induced liver damage. This finding is consistent with the very short doubling time of iCCA reported in the literature (approximately 50 days), which leads to a reduced probability that a secondary biliary cirrhosis develop before the clinical manifestation of iCCA become apparent. We added the following sentence:

Line 123. Notably, no patients had a pattern of fibrosis consistent with a secondary biliary cirrhosis due to a biliary obstruction by the tumour.

  • Were there any differences in the clinicopathological characteristics between the Italian and French cohort? How many cases from each cohort were enrolled in the study? Were the donor liver cases also matched for each institution?

We have specified the following aspects:

  • Line 89: A total of 180 patients with ICCA were included in this study (FIGURE 1), including 92 and 88 patients for the Italian and French centers, respectively.
  • Line 112: The prevalence of NAFLD (38.5 vs 53.1%, p=0.113) and NASH (23.4 vs 21.5% p=0.835) were similar in the Italian and French centers, respectively.
  • Line 163: The median OS was 44.6 months (95%CI 34.7-54.5), with no differences between the Italian and French centers (p=0.715).
  • Line 275: Liver donors cases were also matched for each institution.
  • The authors conclude that "NASH was found in more than 20% of patients who would have been classified as having a healthy liver". Did any of these patients have clinical features of metabolic syndrome (hypertension etc) or liver function tests suggestive of NAFLD/NASH, or were they all diagnosed as NAFLD/NASH post-operatively after examining the background liver? The quoted sentence in the conclusion seems to imply that they were all incidentally discovered after surgery, but I feel that a good proportion of them would have actually been diagnosed with metabolic syndrome or NAFLD pre-operatively. 

Thanks for pointing out our error. We actually meant “NASH was found in more than 20% of patients who would have been classified as having no established risk factors”.  The reviewer is right when feeling that a proportion of patients had been previously diagnosed with NAFLD pre-operatively. Still, this proportion was relatively low, as only 11 patients were diagnosed as NAFLD pre-operatively (and no one with NASH). We feel that this point is interesting and would like to thank the reviewer. We modified the text as follows:

- Line 117: Notably, only 11 iCCA patients (6.1% of the whole population) had a pre-operative medical history of NAFLD and no patients had a diagnosis of NASH..

- Line 363: In particular, NASH was found in more than 20% of patients who would have been classified as having no classical risk factors

6) There are few places with awkward phrasing and typographical errors in the manuscript, for example:

Line 32: "hyperexpressed" in iCCA patients -> changed with “overexpressed”

Line 92: A majority of patients "hade" -> changed with “had”

Line 225: "cause fo death" -> changed with “cause of death”

Line 335: "an healthy liver" -> sentence deleted

                Thanks for pointing out. These errors have been corrected

Reviewer 3 Report

De Lorenzo et al. performed a retrospective study to address whether nonalcoholic fatty liver disease (NAFLD) and non-alcoholic steatohepatitis (NASH) are risk factors for intrahepatic cholangiocarcinoma (iCCA). Their data of Cox regression analysis revealed that not NAFLD but NASH was an independent factor of iCCA. Moreover, iCCA patients with NASH had a significantly shorter survival than those without NASH.

Major points:

  1. The author mentioned “data about NAFLD/NASH and intrahepatic carcinoma (iCCA) are few and contradictory, coming from population registries which do not allow a correct distinction between NAFLD and NASH”. I don’t think this is an objective conclusion. Major associations, like AASLD, EASL, and APASL, have published clear definition and guidelines of NAFLD and NASH for several years. NAFLD and NASH can be correctly diagnosed in many medical centers and health care facilities in the US and in many countries now.

  1. The relationship between NASH and iCCA has been reported (please refer to the following reference: J Surg Oncol. 2016 Jun;113(7):779-83; Hepatology 2019 April;69(4):1803-1815). No novel view of points and mechanical findings are provided in this study. I would like to suggest the author to submit this manuscript to a hepatology-specific journal.

  1. “We also underline the need for prevention protocols for NASH patients, ……….. the pathogenesis of NASH-related iCCA”. The sentence is overstated. As we know that the incidence of cholangiocarcinoma is geographically variable. For people who live in eastern Asia, they currently have the highest incidence of iCCA but not often suffer NASH. There is no evidence of a pathogenic effect of NASH on iCCA unless the mechanisms are shown.

Other comments:

  1. Line 39: should be intrahepatic “cholangio”carcinoma (iCCA).

  1. The gender is not significant in the univariate Cox (p = 0.091), why is it selected into multivariate analysis? A wrong selection of factors may lead to an incorrect result.

  1. It is not surprising to see a poorer prognosis of the NASH-iCCA double positive patients than iCCA patients without NASH. In addition to NASH, analyses of panels of classical risk factors of NASH would be designed aiming to develop optimized prognostic tools.

Author Response

De Lorenzo et al. performed a retrospective study to address whether nonalcoholic fatty liver disease (NAFLD) and non-alcoholic steatohepatitis (NASH) are risk factors for intrahepatic cholangiocarcinoma (iCCA). Their data of Cox regression analysis revealed that not NAFLD but NASH was an independent factor of iCCA. Moreover, iCCA patients with NASH had a significantly shorter survival than those without NASH.

Major points:

  1. The author mentioned “data about NAFLD/NASH and intrahepatic carcinoma (iCCA) are few and contradictory, coming from population registries which do not allow a correct distinction between NAFLD and NASH”. I don’t think this is an objective conclusion. Major associations, like AASLD, EASL, and APASL, have published clear definition and guidelines of NAFLD and NASH for several years. NAFLD and NASH can be correctly diagnosed in many medical centers and health care facilities in the US and in many countries now.

We apologize if our sentence was misleading. We did not mean that NAFLD and NASH were not diagnosed correctly nor that the guidelines have not been followed. Rather, we underlined that all these studies have been performed using the ICD-9 classification and that both NAFLD and NASH are categorized under the same ICD-9 code (571.8): therefore, population registries are not able to discriminate one condition from another. For instance, in the papers by Choi et al, Welzel et al, and Chang et al  (references 4,5 and 6), the Authors searched  large databases  the underlying liver disease was identified researching the pertinent databases for the ICD-9 codes of chronic liver diseases. The code 571.8 identified both NAFLD and NASH, so distinction between the two conditions was not possible. In the case of the paper by Stepien (reference 7), NAFLD was diagnosed using the HSI score, which is a score used to identify the presence of fatty liver but not able to provide discriminatory information between NAFLD and NASH. The creators of the HIS index clearly mention this limitation (PMID: 19766548). Also, it was not in Stepien et al intention to discriminate NASH.

Therefore, we believe that our study provided useful information discriminating one condition for another. However, we fully understand the concern of the reviewer, therefore we added the following specification (red font):

Line 66: “The use of data deriving from population registries in which the diagnosis of NAFLD/NASH was searched using the International Classification of Diseases,Ninth Revision (ICD-9) code 571.8 did not allow a discrimination between the two conditions, being both grouped under the same code and  term (“Other chronic nonalcoholic liver disease”). In a more recent registry study, NAFLD was coded according to the results of the hepatic steatosis index, which can not identify the subgroup of patients with NASH.”

  1. The relationship between NASH and iCCA has been reported (please refer to the following reference: J Surg Oncol. 2016 Jun;113(7):779-83; Hepatology 2019 April;69(4):1803-1815). No novel view of points and mechanical findings are provided in this study. I would like to suggest the author to submit this manuscript to a hepatology-specific journal.

We thank the reviewer for sharing the suggestion of these two papers. The first paper, (Kinoshita et al) was already mentioned in our paper (reference 8). As we mentioned in the Introduction, this paper provided very useful insights but can not be conclusive due the choice of a control group which was not no representative of the general population. In detail, the Authors chose patients with liver metastasis from colorectal cancer as a control group. Unfortunately, metabolic syndrome is also a risk factor for colorectal cancer, so the risk of over-representation of NAFLD and NASH in the control group was not negligible. The second suggested paper is a review of the literature. Despite not providing original findings, this review surely contains important information and was therefore added as reference 12  (as suggested by the Reviewer).

Regarding the suggestion of an hepatology-specific journal, we feel that the topic of this paper might be of interest to a broad readership such as that  of Cancers. Also, most of the previous works on this topic were published in non-hepatology journals (for instance references 6,7,8, and 9).

  1. “We also underline the need for prevention protocols for NASH patients, ……….. the pathogenesis of NASH-related iCCA”. The sentence is overstated. As we know that the incidence of cholangiocarcinoma is geographically variable. For people who live in eastern Asia, they currently have the highest incidence of iCCA but not often suffer NASH. There is no evidence of a pathogenic effect of NASH on iCCA unless the mechanisms are shown.

We agree that different risk factors can be involved in different risk area, even if it should be noted that the incidence and prevalence of NASH and NASH-related hepatocellular carcinoma are rising also in Eastern Asia (reviewed in PMID: 23094755).

We also agree that finding a direct evidence of a pathogenic effect of NASH on iCCA should be of pivotal importance. However, a direct pathogenic link has not been demonstrated for NASH and HCC. Still, this association is now recognized even without conclusive mechanistic studies.

According to the reviewer’s suggestion we have modified the following parts:

Line 34 (lay summary): “Prevention protocols for NASH patients should consider also the risk for iCCA and not only HCC. Mechanistic studies aimed to find a direct pathogenic link between NASH and iCCA could add further relevant information”

Line 53 (abstract): “Prevention protocols for NASH patients should consider also the risk for iCCA and not only HCC. Mechanistic studies aimed to find a direct pathogenic link between NASH and iCCA could add further relevant information”

Line 367 (conclusions): “Effective secondary prevention protocols for NASH patients as well as mechanistic studies to unravel the

pathogenic effects of NASH are needed to improve the outcome of these patients”.

Other comments:

  1. Line 39: should be intrahepatic “cholangio”carcinoma (iCCA).

Thanks for pointing out our error. We corrected the sentence.

  1. The gender is not significant in the univariate Cox (p = 0.091), why is it selected into multivariate analysis? A wrong selection of factors may lead to an incorrect result.
  2.  

We thank the reviewer for pointing out a missing information in the Methods and for underlining the need for a specification of the method sused in the multivariable regression. Indeed, we forgot to mention in the Method session that all variables reaching a p<0.10 were included in the multivariable models. The choice of this threshold  is accepted by multiple key opinion leaders in different medical fields (for instance PMID: 11722062, 24703956). It derives from the fact that it is entirely possible that the effect of the evaluated variable is masked by another variable. In any case, we adopted a backward stepwise multivariable analysis. According to this analysis, all of the selected variables are entered in the model, than the variable with the highest p value is removed from the model at each single step, until there are no variable with a p-value>0.05. Thus even if we had theoretically introduced a variable with a very high p values, this variable would have been eliminated right at the first step of the model. In our analysis, sex was the first variable exiting from the model  To ensure that we addressed correctly the point raised by the reviewer, we created a multivariable model without including gender: the results were not modified (thus confirming that the backward stepwise analysis worked correctly. In conclusion: 1) our selection of factor was correct according to a consolidated trend in literature; 2) even if this trend is not acknowledged and this selection is considered “wrong”, it did not impact to the results and did not lead to incorrect results. We corrected our statement as follows :

Line 349: “Variables reaching a p<0.10 at the univariate analysis were further examined by backward stepwise multivariate analysis”

  1. It is not surprising to see a poorer prognosis of the NASH-iCCA double positive patients than iCCA patients without NASH. In addition to NASH, analyses of panels of classical risk factors of NASH would be designed aiming to develop optimized prognostic tools.

Actually, NASH-iCCA double positive patients were shown to be unrelated to the prognosis in the only previous study about this topic (Reddy et al, reference 9), so we think that our findings can not be considered as unsurprising. The presence of a comorbidity in iCCA patients can act on the survival only if it represents a concurrent cause of death (i.e. if patients can die due to the comorbidity before dying from cancer progression ). Therefore, the presence of a concurrent condition does not automatically imply that it will shorten the survival of an iCCA patient.

The reviewer is absolutely correct when stating that the risk factors of NASH should be analysed: this analysis has been performed in Table 5 and demonstrated that every single risk factor (for instance diabetes and BMI) did not affect the survival, but only NASH did.

Round 2

Reviewer 2 Report

My comments have been sufficiently addressed, and I have no further comments. 

Reviewer 3 Report

The authors have addressed all the issues raised. I have no further comments.